# Decision Making of Agile Patterns in Offshore Software Development Outsourcing: A Fuzzy Logic-Based Analysis

Maryam Kausar [1,]*[], Noushin Mazhar [1], Muhammad Ishtiaq [2][] and Amerah Alabrah [3][]

1   Department of Software Engineering, Foundation University Islamabad, Islamabad 44000, Pakistan
2   Department of Data Science and Artificial Intelligence, National University of Computer and Emerging Sciences, Islamabad 44000, Pakistan
3   Department of Information Systems, College of Computer and Information Sciences, King Saud University, Riyadh 11543, Saudi Arabia
*   Correspondence: maryam.kausar@fui.edu.pk

**Abstract:** Computation intelligence techniques are important for making decisions in an agile-based offshore software development paradigm. Offshore development faces additional challenges, such as trust, communication and coordination, and socio-cultural and knowledge transfer. There is a need to determine the rankings of challenges considering their criticality concerning practitioners working in agile-based offshore software development. This paper aims to identify and rank agile challenges in offshore software development by applying computational intelligence techniques. From the systematic literature review, we identified 30 communication and coordination challenges. The distributed agile pattern catalog consists of 15 patterns, from which eight were used to solve communication and collaboration challenges. Many researchers have used fuzzy logic to quantify their results. We further applied the fuzzy analytical technique to determine the priority order concerning the criticality of the identified agile pattern catalog. The results showed that Central Code Repository Pattern ranked the most significant for solving communication and coordination challenges. Global Scrum Board Pattern and Synchronous Communication Pattern ranked second.

**Keywords:** agile software development; global software development; fuzzy sets; distributed agile patterns; computational intelligence

**MSC:** 03B52

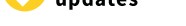

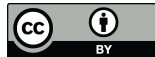

## 1. Introduction

Globalization is progressively taking the IT world. It has affected how software companies do business globally. Software companies adopted the concept of offshore development to gain financial benefits by transferring work to low cost labor forces [1]. Due to the cutting down of costs many companies are opting to offshore their work. Cost reduction has always been the core reason to offshore but there are other advantages too such as access to skilled resources and proximity to markets and customers [2,3]. However, when a team is distributed in offshore ventures, it is often divided by major differences in time, geography, values and business cultures. Trust, socio-cultural, communication, coordination, and knowledge transfer issues emerge as a result [4]. These problems create obstacles for distributed teams, and has an impact on the applicability of agile-based offshore software development, which relies heavily on face-to-face communication [5–7].

Agile Software Development focuses on customer collaboration, changing needs, individual interactions, and continual delivery of working software. Global Software Development using Agile has benefits but it is not a straightforward process, as now the teams are distributed among distinct locations and face-to-face communication is curtailed [8]. To overcome the issues mentioned above, several teams have tried to adapt

agile approaches, but their efforts were solitary and communication with other practitioners challenging.

The proposed study aimed to utilize the computational intelligence technique i.e., fuzzy logic. As distributed agile development involves more complicated uncertainties due to unpredictable ambiguities, probability theory and statistics were used to address the uncertainty. However, in daily situations, natural language is exercised to express thinking and subjective perceptions. In natural language, words might not convey clear meaning. Individual expressions about an event may use the same words but have different subjective perceptions. To tackle this problem, fuzzy set theory is used to express linguistic variables more appropriately [9,10]. Fuzzy logic has been used to evaluate agile methods and practices [11]. Researchers have used fuzzy logic to explore different aspects of agile, such as requirement estimation, risk assessment, and agility evaluation. In this paper, fuzzy logic is used to validate the use of the Distributed Agile Patterns Catalog.

Work has been done on distributed agile development but has only focused on either trust [12] or on communication with remote customers [13]. CMMI [14] and Blockchain [15] has also been used to propose solutions in distributed development, but the work has focused on maturity levels and Ethereum blockchain to execute smart contracts for payment distribution. This research focused on solving the 30 communication and coordination challenges that occur in distributed agile development by applying the Distributed Agile Pattern Catalog with the utilization of fuzzy logic.

The paper is organized as follows. Section 2 discusses the important related research in communication and coordination. Section 3 presents the research methodology used in this research. Section 4 presents the Distributed Agile Patterns and Section 5 explains the application of fuzzy logic to evaluate the usefulness of agile patterns. Section 6 discusses the findings of the paper. Section 7 describes the threat to validity and Section 8 provides the conclusion of the paper, including the theoretical and practical contributions of the proposed study.

## 2. Related Work

Global Software Development (GSD) is a trend that is only just starting to gain momentum and appeals to businesses all around the world. It unites different partners from several nations and cultures to access a vast labor pool. As new businesses go global to reduce their costs, there are unique challenges associated with GSD [16,17]. Among them are the diversity of sourcing and the complexity of communication and coordination that are required to prevent global project failure [18,19]. Agile software development in a distributed scenario is challenging. Study has been done to investigate the impact of the agile framework on IT sourcing and to identify the dimensions of ITS affected by agile frameworks [20]. Some companies use product line engineering methods to shorten their release cycles for distributed agile projects. These changes result in reducing the risks due to communication and coordination, as well as help in sharing knowledge across the team [21]. Work has also been done on the use of technology in distributed development [22–24]. According to the authors if we carefully select technology and allow members to participate, irrespective of their location, we can have efficient distributed development.

Challenges such as trust, communication and coordination, socio-cultural, and knowledge transfer issues, occur due to temporal, geographical and socio-cultural distance affecting communication and coordination in offshoring. Conchuir et al. mapped opportunities and challenges of global software development in terms of communication, coordination, and control. It was seen that increase in distance resulted in an increased cost of communication and coordination, and was highly dependent on synchronous tools, which could result in misunderstandings [25].

Ghani et al. mapped how distance related to difficulties in coordination and communication in a systematic literature review [26]. It was shown that physical distance contributed to 35% of challenges to communication and coordination, and temporal distracts contributed to 28%, socio-cultural to 22%, and knowledge experience to 4%. According to

Niazi [27] offshore projects having closer geographical and time zone proximity allow more communication, as compared to greater distances, which results in fewer communication and coordination challenges. Khan et al. [28,29] identified that intercultural challenges are faced by companies due to differences in language and cultural values that have a negative impact on the communication and coordination processes between clients and vendors.

As businesses began to use offshore, new patterns emerged. Noll et al. [30] designed Decision support system patterns for offshore development, and identified sixty specific practices in the Global Teaming Model which were made accessible to managers and developers involved in offshore development. Similarly, Siemens designed collaboration patterns to help create trust among team members [31]. Van Heesch et al. introduced two distributed collaborative patterns that emphasized enhancing collaboration among distributed teams [32]. Inayat et al. proposed collaborative patterns for agile software development teams that were guided by requirements [33].

Considerable research has been conducted on designing agile offshore patterns. Cordeiro et al. designed organizational patterns with Scrum [34]. Välimäki et al. designed patterns for management techniques in distributed scrum [35]. In distributed agile development, study has also been done on finding patterns for requirements [36]. Kausar et al. compiled a catalog of distributed agile patterns to be used by practitioners opting for offshore development [37]. Existing efforts include using patterns in GSD, as well as using agile practices, to solve offshoring challenges [38]. Ontologies [39] and DevOps [40] for collecting requirements in GSD have also been studied. The research done in this area so far has either been too broad or too narrowly focused on managing offshore projects. The Distributed Agile Pattern catalog, on the other hand, is intended to help practitioners adopt agile practises in an offshore setting.

Artificial intelligence (AI) application and its impact on innovation, revolution, and decision-making have been widely debated in recent years. AI is becoming an integral part for organizational procedures, as it can positively influence decision making [41]. Fuzzy logic has been used for agile. Raslan et al. [42] designed an innovative framework for accurate effort estimation. The proposed framework utilized fuzzy logic, Story Point (SP), Implementation Level Factor (ILF), Friction factor (FR), and Dynamic Force (DF). Similarly, Saini et al. [43] used fuzzy logic in improving effort estimation in agile software development. To estimate effort, they used three input variables: the user story, the team's expertise, and the complexity. Singhal et al. [44] proposed a novel fuzzy logic approach in the Agile Security Framework for risk assessment and threat prioritization. The DREAD model is known to deal well with risks related to security threats, but it allows only crisp values. Other approaches have been used for Distributed Agile Development, such as CMMI and blockchain. In [14], C2M addressed CMMI maturity models that are focused on number of maturity levels, and number of factors and practices, among other aspects. Another existing approach that has been used for distributed agile development is blockchain [15]. The authors used AgilePlus, which is a private Ethereum blockchain, to execute smart contracts for payment distribution among development teams. However, our focus is on the communication and coordination of distributed teams.

Frameworks have been developed for distributed agile teams that focus on developing trust among distributed team members. The framework depends on elements that relate to the workplace, leadership, organizational structure, individual perspective, and socio-cultural perspective. The study in [12] just focused on trust, whereas our research focuses on different aspects related to communication and coordination among distributed teams. Recently, work has been done on coordination of distributed agile teams where the focus is on coordinating with remote customers [13]. Our research provides a patterns approach, focusing on communication and coordination of the development team, and patterns, such as Follow-the–Sun and visiting onshore and offshore, to elaborate on how to interact with remote customers.

The focus of this research is to solve the challenges caused by communication and coordination by using the Distributed Agile Pattern Catalogue with the utilization of fuzzy logic.

## 3. Research Methodology

A thorough literature study was conducted in order to pinpoint the problems with coordination and communication in distributed agile development. This review used the following five well-known electronic databases: Google Scholar, ACM Digital Library, SpringerLink, ElsevierScienceDirect and the IEEE Xplore. These databases contained enough research material for the subject. Moreover, manual searches were done for XP, XP/Agile Universe, and Agile Development Conference. Each stage of the review process is shown in Figure 1. The terms and keywords used for the search in the first stage are listed in Table 1. With the help of the Boolean "AND" operator, all articles on "Communication", "Coordination", "Offshore Software Development", and "Agile Practise" could be found. In other words, we looked for every combination of a single item from each of the first, second, third, and fourth categories. Discussion comments, editorials, news, summaries, reviews, correspondences, conversations, readers' letters, and summaries of tutorials and workshops were not included in the search. A total of 5647 "hits" were generated by this search approach. However, only 4899 hits remained once duplicate papers were taken out of the equation.

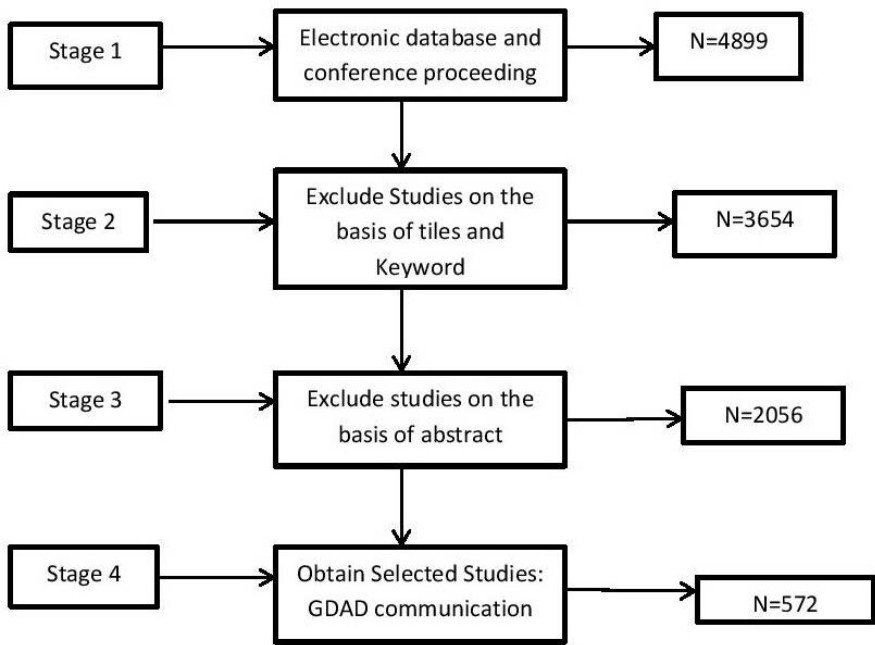

**Figure 1.** Study selection process.

The second stage involved eliminating articles based on how closely the titles and keywords adhered to the search terms listed in Table 1 for each article. In total, 3654 articles were chosen after reading all the titles. After reading all abstracts, and only including publications that demonstrated a connection to the offshore experience, the third stage resulted in the shortlisting of 2056 articles. Lastly, articles that only focused on DAD communication and coordination were selected. In this stage, 572 articles were selected. The selection process is shown in Figure 1.

Based on the finding of the systematic literature review, Table 2 was designed to show the challenges and the impact caused due to the communication and coordination issues in the distributed agile development (DAD) process.

**Table 1.** Search terms.

| Search Category | Keywords |
|---|---|
| Communication | Communication, cross-team communication, team communication, outsourcing communication, offshore communication,communication technology, communication tool. |
| Coordination | Coordination, coordinate teams, coordination tools, coordination technology, offshore coordination. |
| Offshore Software Development | Offshoring Agile, Distributed agile, Offshore Development, global software development, global agile development, global software, offshore software development, distributed development, distributed development teams, global software, development, global development, outsourcing development, engineering, global software engineering. |
| Agile Practices | Agile, agile practice, agile methods, Scrum, Scrum method, Scrum practice, XP, extreme programming, XP method, XP practice |

**Table 2.** Challenges cause due to communication and coordination issues in DAD.

| No. | Category | Challenge |
|---|---|---|
| 1. | Requirements | Misunderstanding of requirements leads to dependency on expert decisions. Lack of meeting minutes to clarify requirements. Misunderstood requirements across sites. Lack of involvement of developers and tests in requirement meetings. |
| 2. | Team Configuration | Lack of task awareness Lack of trust Early communication difficulty Difficulty in team formation Less understanding of teamwork across the sites Difficulty caused due to management of items by different people Problems in knowledge sharing Cultural differences in communication Technical barriers to setup a video conference Lack of required formal documentation. Lack of training. |
| 3. | Customer Communication | Less frequent communication with customers. Hiding information from customers. Lack of feedback. |
| 4. | Meetings | Reduced communication opportunities Lack of face-to-face and informal communication Long-time communication using technology Meeting only at one site |
| 5. | Project Characteristics | Tendency to lose track of the overall work process Difficulty in keeping areas separate Lack continuous integration Lack of process improvement knowledge. |
| 6. | Organizational Factors | Extra cost due to training teams Issues in creating a shared understanding of change requests Difficulty in creating transparency among sites Lack of organizational support. |

To find answers, these problems were mapped onto the list of Distributed Agile Patterns. Using fuzzy logic, the results were validated. A summary of the Distributed Agile Patterns library is provided in Section 4.

## 4. Distributed Agile Patterns

According to [45], a "pattern" is a reusable solution for a reoccurring problem in a certain environment. Agile patterns are defined as "focus on how an agile technique is being frequently adjusted and used in order to tackle a recurrent agile challenge in a given context", based on the definition of patterns provided above [46]. The entire software development life cycle is currently covered by six different types of patterns: Requirement patterns, Analysis patterns, Design patterns, Architecture patterns, Idioms, and Anti-patterns. Similar work has been done on agile patterns, offshore patterns, and software process patterns for agile methodologies [47]. Nevertheless, as indicated in the Related Work section, such efforts have been limited, which is why the Distributed Agile Patterns catalog was developed.

For our catalog, we used Gamma's pattern template to maintain familiarity during customisation, since they are thought to be the first pattern catalog recorded by the software community. A pattern typically contains four fundamental components: pattern name, problem, solution, and consequence. A total of 15 distributed agile patterns were divided into four categories for the catalog: management, communication, collaboration, and verification. The eight distributed agile development patterns built on communication and cooperation are the main topic of this study. We organized all the patterns into categories in Table 3.

Due to the limited space available, only one pattern is presented in this paper as a sample. The full catalog is available on the following URL: http://usir.salford.ac.uk/id/eprint/46308/1/Maryam%20Kausar.pdf (accessed on 8 March 2023).

**Table 3.** Distributed agile patterns catalog.

| | Category | | | |
|---|---|---|---|---|
| | **Management Patterns** | **Communication Patterns** | **Collaboration Patterns** | **Verification Patterns** |
| | Distributed Scrum of Scrum | Global Scrum Board | Collaborative Planning Poker | Project Charter |
| | Local Setup Meeting | Central Code Repository | Follow-the-sun | Onshore Reviews Meetings |
| Pattern Name | Local Sprint Planning | Asynchronous information Transfer | Collective Project Planning | |
| | Local Pair Programming | Synchronous Communication | Visit onshore- offshore | |
| | Asynchronous Retrospectives | | | |

Collaborative Planning Poker Pattern

Playing planning poker allows an agile team to assign point estimates to each story card. This activity involves the product owner as well. The development team assigns an estimate to a story card after learning from him or her about its purpose and worth. The team members who give the lowest and highest estimation explain why, based on the points allotted. Each story is the object of a brief team dialog, and the team then decides on an estimation.

It has been observed that even though the team is distributed during the project planning activity, the planning poker activity is still conducted when both teams are present. This observation led us to identify the following pattern.

Pattern Name

Collaborative Planning Poker Pattern

Intent

Onshore and offshore team members participate in this activity.

Also Known As

Scrum Poker or Planning Poker.

Category

Collaborative category, as this pattern enables the discussion of story card duration between the onshore and offshore teams.

Motivation

Addressing the trust, socio-cultural, communication and coordination, and knowledge transfer difficulties is the driving force behind this design. For instance, before a team can begin working on a project that is spread across multiple time zones, everyone on the team must agree on the length of time for each feature. This makes project progress visible and aids in estimating how long it will take to complete the project. The onshore and offshore team members engage in a game of planning poker to come to an agreement on the estimation of a story card. Once the estimate has been decided, it is written down, accepted by the product owner or customer, and then the next story card is estimated, and so on.

Applicability

The Planning Poker pattern is used when:

- The team is distributed across different temporal zones, and each sprint focuses on a different story card.

Participants

- Product owner/Client.
- Distributed onshore and offshore agile teams.

Collaboration

- The client approves the estimation made by the team members.

Consequences

The following are the advantages and drawbacks of the Planning Poker pattern:

1. It enables the onshore and offshore teams to agree on a story card estimation, assisting the team in determining their team pace. The presence of members from both sites during this exercise aids in overcoming issues with trust and socio-cultural barriers.

2. It presents an estimate of project completion to the product owner or client, assisting in overcoming the obstacles of collaboration, communication, and knowledge transfer.

3. The planning poker may get out of control if there is disagreement among the team members on an estimate on a story card.

Known uses

Throughout Asia, South America, and North America, UShardware has development centers. They used planning poker to estimate their story cards when moving to a distributed agile environment [48].

Related Patterns

The Planning Poker Pattern is frequently used in combination with Collective Project Planning since it works best when the entire team is together. Following that, the estimated story cards are posted on the Global Scrum board for the entire team to see throughout the project.

## 5. Results

Based on the challenges identified in Section 3, the Distributed Agile Pattern catalog was studied to see its application to the communication and coordination challenges. Based on the patterns' motivation, each pattern was mapped onto the challenges. Table 4 presents the mapping of the DAP catalog onto the challenges. The DAP catalog originally had 15 patterns but only eight solve challenges related to communication and coordination.

To validate the results fuzzy logic was used, and participants from the industry were invited to take part in this research. We used one parameter, "usefulness", to check the validity of our patterns, as it covered the subset of applicability. The focus of this study was to determine how useful Distributed Agile Patterns are for communication and coordination challenges.

**Table 4.** Applying DAP to solve communication and coordination issues in DAD.

| No. | Distributed Agile Pattern | Challenge | Solution |
|---|---|---|---|
| 1. | Global Scrum Board | Lack of task awareness | A centralized global scrum board is maintained by all sites, which provides awareness of all the tasks to the team members. |
| | | Less understanding of teamwork across the sites | Global scrum board helps keep all team members updated about each other's tasks and progress, resulting in the team understanding each other's work across sites. |
| | | Difficulty caused due to management of items by different people | The global scrum board serves as a central board for sharing information, everyone has access to it, and this helps in the management of items by different people across different sites. |
| | | Problems in knowledge sharing | Global scrum board helps in providing centralized access to all knowledge that is being shared across the sites. |
| | | Lack of required formal documentation. | With constant updates on the global scrum board, all required information is shared. |
| | | Hiding information from customers | We can give viewing access to the global scrum board to the customers so that they can see the progress of the project. |
| | | Tendency to lose track of the overall work process | A The global scrum board helps in keeping track of the overall work process and project status. |
| | | Issues in creating shared understanding of change requests | As the board is updated regularly it maps all change requests. |
| 2. | Central Code Repository | Less understanding of teamwork across the sites | All sites share a central code repository, which helps each site see each other's progress. |
| | | Difficulty caused due to management of items by different people | Code is shared in a central code repository, which helps in managing work done by different people |
| | | Problems in knowledge sharing | Code repository aids in sharing knowledge of code across sites. |
| | | Lack of required formal documentation. | While sharing code in a central repository, formal documentation is maintained so that each site can understand the work done. |
| | | Tendency to lose track of the overall work process | All code is shared in a central code repository, it helps keep track of the work done. |
| | | Lack continuous integration | The central code repository is updated regularly, which solves the issue of continuous integration problems. |
| 3. | Asynchronous Information Transfer | Lack of meeting minutes to clarify requirements | Asynchronous tools can be used to share meeting minutes. |
| | | Asynchronous Information Transfer | As teams follow the follow-the-sun approach, regular updates are exchanged through asynchronous and synchronous tools. |
| | | Difficulty in team formation | The team members depend highly on asynchronous information transfer through tools, which helps in team formation. |
| | | Lack of training. | Asynchronous tools can be used to train teams. |
| | | Less frequent communication with customers | With the use of asynchronous tools, information is also shared with the customers. |
| | | Hiding information from customers | Customers can use asynchronous tools to inquire about any aspect of the project. |
| | | Lack of feedback | Customers can give continuous feedback using asynchronous tools. |
| | | Reduced communication opportunities | Proper use of asynchronous tools can provide the required communication among sites. |
| | | Lack of face-to-face and informal communication | Asynchronous tools like Slack encourage informal communication. |

**Table 4.** *Cont.*

| No. | Distributed Agile Pattern | Challenge | Solution |
|-----|---------------------------|-----------|----------|
| | | Long-time communication using technology | Initially set agenda through asynchronous tools for meetings before setting up synchronous meetings. This helps in keeping the meeting on-schedule. |
| | | Meeting only at one site | With the use of asynchronous tools, combined meetings can be held across sites, by sorting different time zone issues. |
| | | Lack of process improvement knowledge | Asynchronous tools keep all sites informed about improvements in the project. |
| | | Issues in creating shared understanding of change requests | Through asynchronous tools, the procedure to accept application of the change request can be discussed. |
| | | Difficulty in creating transparency among sites | Asynchronous tools provide regular project updates that result in transparency among the sites. |
| 4. | Synchronous Communication | Misunderstood requirements across sites | As teams follow the follow-the-sun approach, regular updates are exchanged through asynchronous and synchronous tools. |
| | | Difficulty in team formation | Team members continuously update each other's progress through synchronous communication, which encourages team formation. |
| | | Lack of training | Synchronous tools can be used to train teams. |
| | | Less frequent communication with customers | Synchronous tools are used to update the progress of the project to the customers. |
| | | Hiding information from customers | Customers can use synchronous tools to inquire about any aspect of the project. |
| | | Lack of feedback | Customers can give continuous feedback using asynchronous tools. |
| | | Reduced communication opportunities | Proper use of both asynchronous and synchronous tools can provide the required communication among sites. |
| | | Lack of face-to-face and informal communication | With voice and video tools; face-to-face communication gap can be reduced. |
| | | Long-time communication using technology | Initially set agenda through asynchronous tools for meetings before setting up synchronous meetings. This helps in keeping the meeting on-schedule. |
| | | Meeting only at one site | With the use of synchronous tools, combined meetings can be held across sites, by sorting different time zone issues. |
| | | Lack of process improvement knowledge | Synchronous tools keep all sites informed about improvements in the project. |
| | | Extra cost due to training teams | With the help of synchronous tools, online training can be conducted. |
| | | Issues in creating shared understanding of change requests | With the help of synchronous tools, the procedure to accept application of the change request can be discussed. |
| | | Difficulty in creating transparency among sites | Synchronous tools provide regular project updates that result in transparency among the sites |
| 5. | Collective Planning Poker | Early communication difficulty | Due to collaborative planning poker, team members are encouraged to interact with each other, which reduces early stage communication difficulty. |
| | | Difficulty in team formation | This activity also helps the team members to understand each others technical skills, facilitating team formation activity. |
| | | Difficulty in creating transparency among sites | With collaborative planning poker, everyone works together in estimating tasks, which creates transparency of the project timeline. |
| 6. | Follow the sun | Difficulty in keeping areas separate | By using the follow-the-sun approach, each site works according to their own time zone and works on their separate sprint. |
| 7. | Collective Project Planning | Misunderstood requirements leading to dependency on expert decisions | Since the whole team takes part in project planning activity, the chance of misunderstanding of requirements is reduced. |
| | | Misunderstood requirements across sites | When the team is co-located for the planning activity, they divide the tasks among the sites, which helps in removing the chance of misunderstanding requirements among the sites. |
| | | Lack of involvement of developers and tests in requirement meetings | All team members take part in this activity; hence, giving a chance to all developers and testers to collaborate with each other. |
| | | Early communication difficulty | As the whole team works together to develop the project plan, it gives them the opportunity to interact with each other and overcome early stage communication issues. |

**Table 4.** *Cont.*

| No. | Distributed Agile Pattern | Challenge | Solution |
|---|---|---|---|
| | | Difficulty in team formation | As the team meets and interacts during the early stages of planning they get to know each other personally, which aids in team formation activity. |
| | | Difficulty in creating transparency among sites | At the start of the project all team members do the project planning activity together; this helps in creating transparency among sites. |
| 8. | Visit Onshore Offshore | Misunderstood requirements leading to dependency on expert decisions | Since the whole team takes part in project planning activity, the chance of misunderstanding requirements is reduced. |
| | | Lack of trust | Despite this being a costly activity it is required that the onshore and offshore team members visit each other in order to establish trust. |
| | | Cultural difference in communication | By visiting each other's locations, team members can learn about different cultures, which aids in better communication. |
| | | Technical barriers to set-up video conference | While visiting each other, they can select and set up tools to use for communication. |

Before sending the survey, we mailed the research summary and letter of invitation to websites, such as Google, LinkedIn, and Facebook, and software companies. In reply, 120 experts consented to take part in the research to whom we sent the web link. In total 45 completed surveys were received from which 42 were selected after removing duplications. The expertise of the participants varied from having experience in distributed agile development from 1 year to 15+ years. Figure 2 shows an overview of the experiences of the participants in offshore software development.

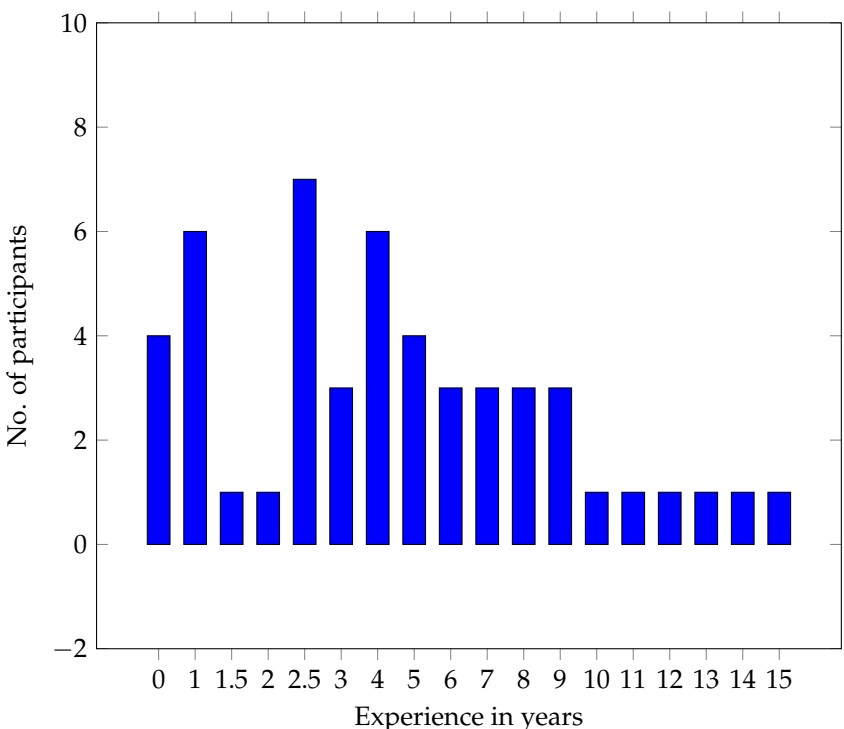

**Figure 2.** Experience of the participants.

*Applying Fuzzy to Rank Patterns*

Fuzzy set theory can be used to replace the subjective fuzziness of human thoughts [49,50]. To demonstrate the understanding of the importance of different patterns, this study introduced a simple method to handle linguistic terms using triangular fuzzy numbers (TFNs). Not all of the patterns are of equal significance which is why we were keen on determining

the importance of weight. A fuzzy set assigns the value of memberships to objects within its universe of discourse in a range of zero to one.

Let U be a universal set, whose elements are u, then, a fuzzy set X is defined by its membership function as follows:

$$\mu_x(u) \rightarrow U[0,1] \tag{1}$$

which allocates to each u a grade of membership X in the interval [0,1].

A linguistic scale was recommended to give actual meaning to understand such situations. We included a seven point linguistic scale to assign the importance weight of patterns, as given in Table 5.

**Table 5.** Linguistic terms for weighting patterns.

| Linguistic Terms | Corresponding Weight of Importance |
|---|---|
| Extremely Agree | (0.9, 1.0, 1.0) |
| Moderately Agree | (0.7, 0.9, 1.0) |
| Slightly Agree | (0.5, 0.7, 0.9) |
| Neutral | (0.3, 0.5, 0.7) |
| Slightly Disagree | (0.1, 0.3, 0.5) |
| Moderately Disagree | (0.0, 0.1, 0.3) |
| Strongly Disagree | (0.0, 0.0, 1.0) |

Similarly, seven linguistic variables, as shown in Table 6, were provided in the survey to rate the implementation of patterns. The technique to obtain the significance weights of patterns is explained in the following steps:

**Table 6.** Linguistic terms for rating patterns.

| Linguistic Terms | Corresponding Weight of Importance |
|---|---|
| Very Useful | (0.9, 1.0, 1.0) |
| Moderately Useful | (0.7, 0.9, 1.0) |
| Marginally Useful | (0.5, 0.7, 0.9) |
| Neutral | (0.3, 0.5, 0.7) |
| Slightly Useless | (0.1, 0.3, 0.5) |
| Moderately Useless | (0.0, 0.1, 0.3) |
| Strongly Useless | (0.0, 0.0, 1.0) |

Step 1: Translate the responses of the survey participant into matrix A using scale, as shown in Table 5. Rows of the matrix indicate the participants and columns correspond to their responses.

$$\mathbf{A} = \begin{bmatrix} \alpha_1^1 & \alpha_1^2 & \alpha_1^3 & \cdots & \alpha_1^n \\ \alpha_2^1 & \alpha_2^2 & \alpha_2^3 & \cdots & \alpha_2^n \\ \alpha_3^1 & \alpha_3^2 & \alpha_3^3 & \cdots & \alpha_3^n \\ \vdots & \vdots & \vdots & \vdots & \vdots \\ \alpha_p^1 & \alpha_p^2 & \alpha_p^3 & \cdots & \alpha_p^n \end{bmatrix} \tag{2}$$

where $p$ represents the total number of patterns and $n$ represents total number of respondents, $\alpha_p^n = \left( l\alpha_p^n, m\alpha_p^n, u\alpha_p^n \right)$ shows the fuzzy weight of the pattern given by the $n$th respondent for $p$th pattern. One example of the result is given in Table 7.

Step 2: The subjective evaluation of each participant varied concerning their experience, role, perception, and understanding of the subject matter. Therefore, we incorporated

the mean score approach to aggregate the fuzzy importance of each pattern by $n$ respondent.

$$w_p = \frac{1}{n} \left[ \sum_{i=1}^{n} \alpha_p^i \right]$$

(3)

where $w_p = (lw_p, mw_p, uw_p)$ shows the aggregate fuzzy importance weight of the $p$th pattern.

Step 3: The aggregated $TFN_{wp}$ was used to obtain the best non-fuzzy performance (BNF) value, $BNP_{wp}$. $BNP_{wp}$ can be produced using Equation (4).

$$BNP_{wp} = \frac{\left[ (uw_p - lw_p) + (mw_p - lw_p) \right]}{3} + lw_p$$

(4)

Here, $BNP_{wp}$ represents the BNP value for the $TFN_{wp}$ while $w_p$ is the important weight of the $p$th pattern in classical (crisp) number format.

Step 4: After the defuzzification of TFN in step 4, crisp numbers were obtained and normalized using Equation (5).

$$R_p = \frac{w_p}{\sum_{p=1}^{n} w_p}$$

(5)

where $R_p$ shows the normalized significance weight of the $p$th pattern, such that $\sum_{p=1}^{n} R_p = 1$.

**Table 7.** Corresponding TFNs(Weighting) of Patterns.

| Experts | Corresponding TFNs( Weighting) of Patterns | | | | | | | |
|---|---|---|---|---|---|---|---|---|
| | **P1** | **P2** | **P3** | **P4** | **P5** | **P6** | **P7** | **B** |
| E1 | (0.7, 0.9, 1.0) | (0.7, 0.9, 1.0) | (0.1, 0.3, 0.5) | (0.7, 0.9, 1.0) | (0.1, 0.3, 0.5) | (0.7, 0.9, 1.0) | (0.9, 1.0, 1.0) | (0.9, 1.0, 1.0) |
| E2 | (0.9, 1.0, 1.0) | (0.9, 1.0, 1.0) | (0.9, 1.0, 1.0) | (0.9, 1.0, 1.0) | (0.9, 1.0, 1.0) | (0.9, 1.0, 1.0) | (0.9, 1.0, 1.0) | (0.9, 1.0, 1.0) |
| E3 | (0.9, 1.0, 1.0) | (0.9, 1.0, 1.0) | (0.7, 0.9, 1.0) | (0.5, 0.7, 0.9) | (0.5, 0.7, 0.9) | (0.7, 0.9, 1.0) | (0.7, 0.9, 1.0) | (0.9, 1.0, 1.0) |
| E4 | (0.7, 0.9, 1.0) | (0.7, 0.9, 1.0) | (0.7, 0.9, 1.0) | (0.7, 0.9, 1.0) | (0.3, 0.5, 0.7) | (0.3, 0.5, 0.7) | (0.3, 0.5, 0.7) | (0.5, 0.7, 0.9) |
| E5 | (0.7, 0.9, 1.0) | (0.7, 0.9, 1.0) | (0.5, 0.7, 0.9) | (0.9, 1.0, 1.0) | (0.7, 0.9, 1.0) | (0.7, 0.9, 1.0) | (0.1, 0.3, 0.5) | (0.5, 0.7, 0.9) |
| E6 | (0.7, 0.9, 1.0) | (0.9, 1.0, 1.0) | (0.7, 0.9, 1.0) | (0.9, 1.0, 1.0) | (0.5, 0.7, 0.9) | (0.7, 0.9, 1.0) | (0.7, 0.9, 1.0) | (0.5, 0.7, 0.9) |
| E7 | (0.3, 0.5, 0.7) | (0.9, 1.0, 1.0) | (0.9, 1.0, 1.0) | (0.9, 1.0, 1.0) | (0.7, 0.9, 1.0) | (0.5, 0.7, 0.9) | (0.9, 1.0, 1.0) | (0.9, 1.0, 1.0) |
| E8 | (0.7, 0.9, 1.0) | (0.9, 1.0, 1.0) | (0.5, 0.7, 0.9) | (0.7, 0.9, 1.0) | (0.7, 0.9, 1.0) | (0.5, 0.7, 0.9) | (0.5, 0.7, 0.9) | (0.9, 1.0, 1.0) |
| E9 | (0.7, 0.9, 1.0) | (0.9, 1.0, 1.0) | (0.9, 1.0, 1.0) | (0.7, 0.9, 1.0) | (0.5, 0.7, 0.9) | (0.3, 0.5, 0.7) | (0.5, 0.7, 0.9) | (0.7, 0.9, 1.0) |
| E10 | (0.7, 0.9, 1.0) | (0.9, 1.0, 1.0) | (0.1, 0.3, 0.5) | (0.9, 1.0, 1.0) | (0.7, 0.9, 1.0) | (0.7, 0.9, 1.0) | (0.1, 0.3, 0.5) | (0.0, 0.1, 0.3) |
| E11 | (0.9, 1.0, 1.0) | (0.7, 0.9, 1.0) | (0.3, 0.5, 0.7) | (0.9, 1.0, 1.0) | (0.9, 1.0, 1.0) | (0.9, 1.0, 1.0) | (0.9, 1.0, 1.0) | (0.9, 1.0, 1.0) |
| E12 | (0.9, 1.0, 1.0) | (0.9, 1.0, 1.0) | (0.9, 1.0, 1.0) | (0.9, 1.0, 1.0) | (0.9, 1.0, 1.0) | (0.9, 1.0, 1.0) | (0.0, 0.1, 0.3) | (0.9, 1.0, 1.0) |
| E13 | (0.7, 0.9, 1.0) | (0.7, 0.9, 1.0) | (0.7, 0.9, 1.0) | (0.7, 0.9, 1.0) | (0.7, 0.9, 1.0) | (0.7, 0.9, 1.0) | (0.7, 0.9, 1.0) | (0.7, 0.9, 1.0) |
| E14 | (0.7, 0.9, 1.0) | (0.9, 1.0, 1.0) | (0.5, 0.7, 0.9) | (0.7, 0.9, 1.0) | (0.9, 1.0, 1.0) | (0.7, 0.9, 1.0) | (0.9, 1.0, 1.0) | (0.3, 0.5, 0.7) |
| E15 | (0.9, 1.0, 1.0) | (0.7, 0.9, 1.0) | (0.5, 0.7, 0.9) | (0.9, 1.0, 1.0) | (0.9, 1.0, 1.0) | (0.7, 0.9, 1.0) | (0.9, 1.0, 1.0) | (0.7, 0.9, 1.0) |
| E16 | (0.9, 1.0, 1.0) | (0.9, 1.0, 1.0) | (0.9, 1.0, 1.0) | (0.7, 0.9, 1.0) | (0.5, 0.7, 0.9) | (0.9, 1.0, 1.0) | (0.7, 0.9, 1.0) | (0.9, 1.0, 1.0) |
| E17 | (0.7, 0.9, 1.0) | (0.9, 1.0, 1.0) | (0.5, 0.7, 0.9) | (0.7, 0.9, 1.0) | (0.9, 1.0, 1.0) | (0.7, 0.9, 1.0) | (0.9, 1.0, 1.0) | (0.9, 1.0, 1.0) |
| E18 | (0.7, 0.9, 1.0) | (0.7, 0.9, 1.0) | (0.7, 0.9, 1.0) | (0.7, 0.9, 1.0) | (0.7, 0.9, 1.0) | (0.7, 0.9, 1.0) | (0.7, 0.9, 1.0) | (0.7, 0.9, 1.0) |
| E19 | (0.9, 1.0, 1.0) | (0.9, 1.0, 1.0) | (0.1, 0.3, 0.5) | (0.5, 0.7, 0.9) | (0.5, 0.7, 0.9) | (0.9, 1.0, 1.0) | (0.7, 0.9, 1.0) | (0.5, 0.7, 0.9) |
| E20 | (0.7, 0.9, 1.0) | (0.9, 1.0, 1.0) | (0.1, 0.3, 0.5) | (0.3, 0.5, 0.7) | (0.9, 1.0, 1.0) | (0.3, 0.5, 0.7) | (0.3, 0.5, 0.7) | (0.7, 0.9, 1.0) |
| E21 | (0.7, 0.9, 1.0) | (0.7, 0.9, 1.0) | (0.5, 0.7, 0.9) | (0.7, 0.9, 1.0) | (0.5, 0.7, 0.9) | (0.5, 0.7, 0.9) | (0.7, 0.9, 1.0) | (0.5, 0.7, 0.9) |
| E22 | (0.3, 0.5, 0.7) | (0.9, 1.0, 1.0) | (0.3, 0.5, 0.7) | (0.3, 0.5, 0.7) | (0.7, 0.9, 1.0) | (0.1, 0.3, 0.5) | (0.0, 0.1, 0.3) | (0.0, 0.1, 0.3) |
| E23 | (0.9, 1.0, 1.0) | (0.9, 1.0, 1.0) | (0.7, 0.9, 1.0) | (0.9, 1.0, 1.0) | (0.5, 0.7, 0.9) | (0.7, 0.9, 1.0) | (0.5, 0.7, 0.9) | (0.7, 0.9, 1.0) |
| E24 | (0.3, 0.5, 0.7) | (0.7, 0.9, 1.0) | (0.3, 0.5, 0.7) | (0.5, 0.7, 0.9) | (0.5, 0.7, 0.9) | (0.5, 0.7, 0.9) | (0.7, 0.9, 1.0) | (0.9, 1.0, 1.0) |
| E25 | (0.7, 0.9, 1.0) | (0.9, 1.0, 1.0) | (0.1, 0.3, 0.5) | (0.9, 1.0, 1.0) | (0.7, 0.9, 1.0) | (0.1, 0.3, 0.5) | (0.7, 0.9, 1.0) | (0.3, 0.5, 0.7) |
| E26 | (0.7, 0.9, 1.0) | (0.7, 0.9, 1.0) | (0.7, 0.9, 1.0) | (0.7, 0.9, 1.0) | (0.7, 0.9, 1.0) | (0.7, 0.9, 1.0) | (0.7, 0.9, 1.0) | (0.7, 0.9, 1.0) |
| E27 | (0.9, 1.0, 1.0) | (0.9, 1.0, 1.0) | (0.7, 0.9, 1.0) | (0.9, 1.0, 1.0) | (0.9, 1.0, 1.0) | (0.7, 0.9, 1.0) | (0.9, 1.0, 1.0) | (0.9, 1.0, 1.0) |
| E28 | (0.9, 1.0, 1.0) | (0.9, 1.0, 1.0) | (0.5, 0.7, 0.9) | (0.7, 0.9, 1.0) | (0.7, 0.9, 1.0) | (0.3, 0.5, 0.7) | (0.9, 1.0, 1.0) | (0.9, 1.0, 1.0) |
| E29 | (0.5, 0.7, 0.9) | (0.9, 1.0, 1.0) | (0.5, 0.7, 0.9) | (0.7, 0.9, 1.0) | (0.9, 1.0, 1.0) | (0.1, 0.3, 0.5) | (0.3, 0.5, 0.7) | (0.9, 1.0, 1.0) |
| E30 | (0.9, 1.0, 1.0) | (0.9, 1.0, 1.0) | (0.5, 0.7, 0.9) | (0.7, 0.9, 1.0) | (0.7, 0.9, 1.0) | (0.7, 0.9, 1.0) | (0.7, 0.9, 1.0) | (0.9, 1.0, 1.0) |

**Table 7.** *Cont.*

| Experts | Corresponding TFNs( Weighting) of Patterns | | | | | | | |
| --- | --- | --- | --- | --- | --- | --- | --- | --- |
| | P1 | P2 | P3 | P4 | P5 | P6 | P7 | B |
| E31 | (0.9, 1.0, 1.0) | (0.7, 0.9, 1.0) | (0.9, 1.0, 1.0) | (0.7, 0.9, 1.0) | (0.7, 0.9, 1.0) | (0.7, 0.9, 1.0) | (0.7, 0.9, 1.0) | (0.7, 0.9, 1.0) |
| E32 | (0.9, 1.0, 1.0) | (0.9, 1.0, 1.0) | (0.9, 1.0, 1.0) | (0.7, 0.9, 1.0) | (0.7, 0.9, 1.0) | (0.7, 0.9, 1.0) | (0.7, 0.9, 1.0) | (0.7, 0.9, 1.0) |
| E33 | (0.9, 1.0, 1.0) | (0.9, 1.0, 1.0) | (0.7, 0.9, 1.0) | (0.9, 1.0, 1.0) | (0.3, 0.5, 0.7) | (0.7, 0.9, 1.0) | (0.5, 0.7, 0.9) | (0.9, 1.0, 1.0) |
| E34 | (0.7, 0.9, 1.0) | (0.9, 1.0, 1.0) | (0.7, 0.9, 1.0) | (0.7, 0.9, 1.0) | (0.7, 0.9, 1.0) | (0.5, 0.7, 0.9) | (0.1, 0.3, 0.5) | (0.1, 0.3, 0.5) |
| E35 | (0.9, 1.0, 1.0) | (0.9, 1.0, 1.0) | (0.7, 0.9, 1.0) | (0.9, 1.0, 1.0) | (0.7, 0.9, 1.0) | (0.7, 0.9, 1.0) | (0.7, 0.9, 1.0) | (0.7, 0.9, 1.0) |
| E36 | (0.7, 0.9, 1.0) | (0.9, 1.0, 1.0) | (0.7, 0.9, 1.0) | (0.9, 1.0, 1.0) | (0.9, 1.0, 1.0) | (0.9, 1.0, 1.0) | (0.3, 0.5, 0.7) | (0.3, 0.5, 0.7) |
| E37 | (0.9, 1.0, 1.0) | (0.3, 0.5, 0.7) | (0.7, 0.9, 1.0) | (0.9, 1.0, 1.0) | (0.3, 0.5, 0.7) | (0.9, 1.0, 1.0) | (0.0, 0.1, 0.3) | (0.7, 0.9, 1.0) |
| E38 | (0.7, 0.9, 1.0) | (0.7, 0.9, 1.0) | (0.5, 0.7, 0.9) | (0.7, 0.9, 1.0) | (0.7, 0.9, 1.0) | (0.5, 0.7, 0.9) | (0.7, 0.9, 1.0) | (0.7, 0.9, 1.0) |
| E39 | (0.9, 1.0, 1.0) | (0.9, 1.0, 1.0) | (0.0, 0.0, 1.0) | (0.9, 1.0, 1.0) | (0.9, 1.0, 1.0) | (0.7, 0.9, 1.0) | (0.9, 1.0, 1.0) | (0.9, 1.0, 1.0) |
| E40 | (0.5, 0.7, 0.9) | (0.7, 0.9, 1.0) | (0.0, 0.1, 0.3) | (0.7, 0.9, 1.0) | (0.9, 1.0, 1.0) | (0.5, 0.7, 0.9) | (0.3, 0.5, 0.7) | (0.7, 0.9, 1.0) |
| E41 | (0.7, 0.9, 1.0) | (0.9, 1.0, 1.0) | (0.7, 0.9, 1.0) | (0.7, 0.9, 1.0) | (0.5, 0.7, 0.9) | (0.7, 0.9, 1.0) | (0.9, 1.0, 1.0) | (0.9, 1.0, 1.0) |
| E42 | (0.9, 1.0, 1.0) | (0.9, 1.0, 1.0) | (0.7, 0.9, 1.0) | (0.7, 0.9, 1.0) | (0.9, 1.0, 1.0) | (0.7, 0.9, 1.0) | (0.9, 1.0, 1.0) | (0.7, 0.9, 1.0) |
| wj | (0.7, 0.85, 0.97) | (0.82, 0.95, 0.99) | (0.57, 0.72, 0.87) | (0.74, 0.90, 0.97) | (0.67, 0.84, 0.94) | (0.61, 0.80, 0.91) | (0.59, 0.76, 0.86) | (0.68, 0.83, 0.92) |

## 6. Discussion

In the research, the Distributed Agile Patterns catalog was used to solve the communication and coordination challenges in offshore development. The reason for selecting this catalog was that it has been previously used to support the requirement engineering process [45]. Based on the findings, we applied it to communication and coordination challenges for agile offshore development.

The DAP catalog was discussed with experts, through which eight patterns were identified as being useful for solving communication and coordination challenges. These eight patterns were mapped onto the 30 communication and coordination challenges that were identified through the literature. To validate the results, experts ranked the usefulness of each pattern to solve the challenges with the help of fuzzy logic. Tables 7 and 8 show the results of the application of fuzzy logic.

**Table 8.** Possible Ranking of the Patterns.

| Pattern # | wj = (l,m,u) | | | BNP_WJ | Rj | Over All Ranking |
|---|---|---|---|---|---|---|
| P1 | 0.75 | 0.9 | 0.97 | 0.873 | 0.13327 | 2 |
| P2 | 0.82 | 0.96 | 0.99 | 0.923 | 0.14090 | 1 |
| P3 | 0.55 | 0.73 | 0.88 | 0.720 | 0.10987 | 8 |
| P4 | 0.74 | 0.9 | 0.98 | 0.873 | 0.13327 | 3 |
| P5 | 0.68 | 0.85 | 0.95 | 0.827 | 0.12614 | 4 |
| P6 | 0.62 | 0.8 | 0.92 | 0.780 | 0.11902 | 6 |
| P7 | 0.6 | 0.76 | 0.87 | 0.743 | 0.11343 | 7 |
| P8 | 0.68 | 0.84 | 0.92 | 0.813 | 0.12411 | 5 |

P2 Central Code Repository Pattern was ranked 1 in terms of usefulness as companies need to invest in a centralized tool to help them synchronize their work across different work locations. This aids in solving all the management and knowledge-sharing issues. In all, 97.8% of the experts agreed that the Central Code Repository pattern facilitates continuous integration and, with regular maintenance, the code quality improves. Of the respondents, 2.2% had neutral opinions.

P1 Global Scrum Board Pattern was ranked at 2 as it solves challenges such as lack of awareness, sharing of formal documentation, and creating a shared understanding of change requests. In all, 88% of practitioners agreed on the usefulness of the patterns. However, 12% believed it to be marginally useful, as designing a global scrum board could introduce scope creeps and extra effort would be required to make detailed burndown charts and retrospective notes.

P4 Synchronous Communication Pattern was ranked 3 in global software development as most communication has moved online, and synchronous tools play an important role, from discussing requirements to collecting feedback. In all, 88% of practitioners agreed that they highly depend on synchronous tools for their day-to-day tasks. However, 12% suggested that although they believe synchronous tools are important they face difficulty in selecting which tool to use for different tasks.

P5 Collective Planning Poker Pattern was ranked at 4 as it helps in the formation of the team and creates transparency early on among the team members at different locations. In all, 68% of the practitioners agreed that teams should perform planning poker activity together as it allows the teams to correctly estimate the work. Although 28% believed that the activity was useful they did not believe they had to be present at the same location, and could rather use online tools to cut down on traveling costs.

P8 Visit Onshore Offshore Pattern was ranked 5 as this practice can help the team members to understand each other's culture and build trust among the team. In all, 73% of the practitioners believed this to be effective, although 10% argued that it is a bad practice as it could be difficult to organize and would cost the team a lot on travel.

P6 Follow-the-Sun Pattern was ranked 6 as it helps distributed teams work according to their own country's working hours and each team can share their progress using a

shared code repository and scrum boards. In all, 68% of practitioners agreed this is a useful pattern, while 10% held the opinion that it was a weak pattern as there are still companies in Pakistan that synchronize their working hours with their onshore teams in the USA and the UK.

P7 Collective Project Planning was ranked 7, as, even though, ideally, it would be better for teams to co-plan the project, most of the planning is done on the onshore site. In all 55% of the practitioners agreed that this practice is useful but 45% disagreed as most of the work outsourced in Pakistan is based on the planning of the onshore site.

P3 Asynchronous Information Transfer was ranked last as even though the teams are distributed, they prefer real-time communication and that is why they highly depend on synchronous tools. In all, 47% of practitioners agreed that asynchronous tools are useful as when you cannot have synchronous communication due to time difference, they can still transfer the information, although 53% preferred synchronous communication.

## 7. Threat to Validity

There exist four primary types of threat affecting four types of validity: internal validity, external validity, construct validity and conclusion validity. We assessed our results by applying threat to the validity. The research concentrated only on board areas such as requirements, team configuration, customer communication, meetings, project characteristics, and organizational factors, while identifying the fundamental inherent issues of communication and coordination. Since their occurrence in literature was not particularly noteworthy, we chose not to concentrate on problems that did not fit within these categories. This could cause threat to the validity because new difficulties might arise that do not fit within existing categories.

The selection of research for the systematic literature review was made based on a list of precise keywords, which could risk a threat to their validity by leaving out some studies, since the keywords did not apply to them. The screening process was carried out manually by reading the papers and selecting 572 on the basis of those that met the three screening criteria. As this process was carried out manually, there is a risk to the validity of the results because researchers' understanding of the papers can vary, and they run the risk of making mistakes or being biased.

A small number of respondents were chosen for the sample set of the questionnaire. This could risk a threat to validity because the choice of research participants was constrained by the availability and willingness of specialists in the field of agile practices. A huge variety of variables can have an impact on the project, just like they can with any empirical software engineering project.

It is challenging to pinpoint single criteria that determine whether or not the distributed agile patterns catalog is successful. The results of the utilization of fuzzy logic, however, made the catalog's utility clear. In a specific context, patterns typically relate to generalized solutions. We make no claims that the catalog is all-inclusive. In order to address offshore challenges, we wish to encourage researchers to find more recurrent agile practices.

## 8. Conclusions and Future Work

Communication and coordination have been widely recognised as key challenges of offshore development, which makes the adoption of agile practices difficult. We mapped eight Distributed Agile Patterns to solve the challenges caused by the communication and coordination issue and ranked them using fuzzy logic. P2 Central Code Repository Pattern was ranked 1, as 97.8% of participants agreed that they need to have a central code repository to solve communication and coordination challenges. Experts ranked the patterns, based on usefulness, from P1 to P8. The percentages varied from 97.8% to 47%, with P3 Asynchronous Information Transfer being the least useful.

Practitioners can use these patterns for their offshore projects. Since patterns are generalized, it is easy for practitioners to adapt them into their own projects and, based on

the fuzzy logic rank, select which pattern is more useful for them according to their project. It is very practicable to use them in real-time, as all practitioners need to do is understand the pattern catalog and implement it into their projects.The pattern catalog is very easy to understand as it is based on similar pattern guidelines of Design Patterns, proposed by Gamma, which everyone in the domain of Software Engineering is familiar with.

*8.1. Theoretical and Practical Contributions*

The suggested research contributes to the discipline of Software Engineering in academia by highlighting the effects of coordination and communication challenges in distributed software development projects. It further helps to clarify the significance of non-technical aspects of distributed development, such as team coordination and communication. The Distributed Agile Patterns catalog presents a guideline for any team wanting to opt for distributed agile development.

The proposed catalog, as seen from the perspective of the software industry and the experts engaged in it, aims to help organisations to greatly improve distributed team coordination and communication. In this regard, we might list a few specific implications for the industry: (i) Establishing a new approach, which focuses on the importance of the communication and coordination process of the project; (ii) Using the catalog to map best practices for improving communication and coordination in the distributed agile development; (iii) Improving collaboration among teams; (iv) Limiting errors through proper planning for the communication and coordination process; (v) Developing agile and effective of distributed projects.

*8.2. Future Work*

Future work aims is to study how organisations will adopt these patterns onto their live projects and to compare the results to see how the catalog helps solve communication and coordination challenges and what the impact is on project success. Currently, the Distributed Agile Patterns catalog has 15 patters, which can be extended to not only focus on communication and coordination challenges, so we can see its impact on other offshore challenges. In addition, another possibility is to develop a computational tool to support the proposed catalog, made available for every organization that wants to use it.

**Author Contributions:** Conceptualization, M.K.; writing—original draft preparation, M.K., N.M., M.I. and A.A.; and writing—review and editing, M.K., N.M., M.I. and A.A. All authors have read and agreed to the published version of the manuscript.

**Funding:** Researchers Supporting Project number (RSP2023R476), King Saud University, Riyadh, Saudi Arabia.

**Institutional Review Board Statement:** Not applicable.

**Informed Consent Statement:** Not applicable.

**Data Availability Statement:** Not applicable.

**Acknowledgments:** The authors would like to acknowledge the Researchers Supporting Project number (RSP2023R476), King Saud University, Riyadh, Saudi Arabia.

**Conflicts of Interest:** There is no conflict of interest.

## Abbreviations

The following abbreviations are used in this manuscript:

| | |
|---|---|
| GSD | Global Software Development |
| DAP | Distributed Agile Patterns |
| DAD | Distributed Agile Development |
| TFN | Triangular Fuzzy Number |
| BNP | Best Non Fuzzy Performance |

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
