# Peer review of "Decision Making of Agile Patterns in Offshore Software Development Outsourcing: A Fuzzy Logic-Based Analysis"

_axioms, doi:10.3390/axioms12030307_

Round 1

Reviewer 1 Report

1.    Introduction is too short, include findings from the prior literature to show research gap develop study objective. 

2.    How practicable is your proposed model in real-time?

3.    What are the contributions of the study (practical and theoretical).

4.    What is the future scope of the study? Explanation in three lines seems not enough.  

5.    Consider the following related papers to embellish your paper:

a.     Šmite, Darja, Claes Wohlin, Aybüke Aurum, Ronald Jabangwe, and Emil Numminen. "Offshore insourcing in software development: Structuring the decision-making process." Journal of systems and software 86, no. 4 (2013): 1054-1067.

b.    Bokhari, Syed Asad A., and Seunghwan Myeong. "Use of artificial intelligence in smart cities for smart decision-making: A social innovation perspective." Sustainability 14, no. 2 (2022): 620.

c.     Amiri, Fouad, Sietse Overbeek, Gerard Wagenaar, and Christoph Johann Stettina. "Reconciling agile frameworks with IT sourcing through an IT sourcing dimensions map and structured decision-making." Information Systems and e-Business Management 19, no. 4 (2021): 1113-1142.

Author Response

Reply to the reviewer comments have been attached.

Reviewer 2 Report

Paper review titled "Decision making of agile patterns in offshore software development outsourcing: A Fuzzy Logic based analysis"

Summary: The authors propose to use the fuzzy logic to quantitatively evaluate the decision of agile pattern inclusion in the catalog for offshore software development scenario.

Comments:

1. Justify the use of fuzzy logic over an alternative approach to the similar problem.

2. Overall the article is well written with few exceptions, specially spacing and typing mistakes may be corrected. 

3. Usefulness of the method has been demonstrated through the obtained results, however, no comparison with the existing methods is provided, please provide the rational for omitting the comparison with any of the similar approaches.

4. A deep proofread is required to improve the readability and understandability of the paper. 

5. I suggest to consider latest refers to support the novelty of this study. 

Author Response

Reply to the reviewer comments have been attached

Round 2

Reviewer 1 Report

The manuscript has been improved perfectly following the given comments. I would suggest to accept this paper please.